# Cost-effectiveness and budget impact analysis of facility-based screening and treatment of hepatitis C in Punjab state of India

Yashika Chugh,[1] Madhumita Premkumar [ID],[2] Gagandeep Singh Grover,[3] Radha K Dhiman,[4] Yot Teerawattananon,[5,6] Shankar Prinja [ID] [7]

For numbered affiliations see end of article.

**Correspondence to**
Dr Shankar Prinja;
shankarprinja@gmail.com

## ABSTRACT

**Objective** Despite treatment availability, chronic hepatitis C virus (HCV) public health burden is rising in India due to lack of timely diagnosis. Therefore, we aim to assess incremental cost per quality-adjusted life year (QALY) for one-time universal screening followed by treatment of people infected with HCV as compared with a no screening policy in Punjab, India.

**Study design** Decision tree integrated with Markov model was developed to simulate disease progression. A societal perspective and a 3% annual discount rate were considered to assess incremental cost per QALY gained. In addition, budgetary impact was also assessed with a payer's perspective and time horizon of 5 years.

**Study setting** Screening services were assumed to be delivered as a facility-based intervention where active screening for HCV cases would be performed at 22 district hospitals in the state of Punjab, which will act as integrated testing as well as treatment sites for HCV.

**Intervention** Two intervention scenarios were compared with no universal screening and treatment (routine care). Scenario I—screening with ELISA followed by confirmatory HCV-RNA quantification and treatment. Scenario II—screening with rapid diagnostic test (RDT) kit followed by confirmatory HCV-RNA quantification and treatment.

**Primary and secondary outcome measures** Lifetime costs; life years and QALY gained; and incremental cost-effectiveness ratio for each of the above-mentioned intervention scenario as compared with the routine care.

**Results** Screening with ELISA and RDT, respectively, results in a gain of 0.028 (0.008 to 0.06) and 0.027 (0.008 to 0.061) QALY per person with costs decreased by −1810 Indian rupees (−3376 to −867) and −1812 Indian rupees (−3468 to −850) when compared with no screening. One-time universal screening of all those ≥18 years at a base coverage of 30%, with ELISA and RDT, would cost 8.5 and 8.3 times more, respectively, when compared with screening the age group of the cohort 40–45 years old.

**Conclusion** One-time universal screening followed by HCV treatment is a dominant strategy as compared with no screening. However, budget impact of screening of all ≥18-year-old people seems unsustainable. Thus, in view of findings from both cost-effectiveness and budget impact, we recommend beginning with screening the age cohort with RDT around mean age of disease presentation, that is, 40–45 years, instead of all ≥18-year-old people.

### Strengths and limitations of this study

► Our analysis is the first state-level study in India where the country population contributes to around 7.5% of the global disease prevalence.
► We used a biologically plausible model which is fitted with local data on disease epidemiology and effectiveness of directly acting antiviral agents from Punjab, India.
► Cost of treatment is drawn from the National Health System Cost Database and nationally representative household surveys for patient costs.
► We did not consider the possibility of reinfection once treated in our model.
► We acknowledge that the sample sizes for estimating the utility values may not be large enough for different health states.

## INTRODUCTION

Nearly 71 million people are living with hepatitis C virus (HCV) infection around the globe.[1 2] HCV infection led to around 399 000 deaths mainly caused by cirrhosis and hepatocellular carcinoma (HCC).[2] With the advent of all-oral directly acting antiviral agents (DAAs), it is possible to cure 90% of the people infected with HCV; however, the awareness of infection, access to diagnostics and treatment facilities are low.[2 3] Globally, only one in five people living with HCV infection had been diagnosed.[4] Further, among those infected with chronic HCV, only 1% of them have accessed the newer antiviral-based treatment.[5]

The World Health Assembly released the global targets to eliminate viral hepatitis by 2030.[6] The aim is to diagnose 90% of the hepatitis B virus (HBV) and HCV infections, and treat 80% of the eligible infections by 2030.[6 7] With around 40% of the HCV-infected people residing in low/middle-income countries (LMICs), India constitutes the fourth

highest number of population with viremic HCV, where only less than 1% of people are aware of their infection status.[4 8 9]

Though the national prevalence of chronic HCV infection in India ranges from 0.9% to 1.9%,[10 11] it is as high as 3.6% in the northern state of Punjab.[12 13] The high prevalence in Punjab can be attributed to the high percentage of unsafe medical procedures and blood transfusions.[12] Furthermore, Punjab ranks second among the top 10 Indian states with highest number of people who inject drugs (PWID), which is a contributor to increased risk for reinfection as well.[12 14] However, it is difficult to identify such people due to associated stigma.

In 2016, Punjab government launched the 'Mukh Mantri Punjab Hepatitis C Relief Fund' (MMPHCRF) for free treatment including DAAs as well as diagnostic and monitoring tests required during the treatment of patients with HCV.[15] Despite free treatment, due to lack of awareness of disease status, less than 10% of the total estimated cases in Punjab have been put on treatment.[16] Thus, along with implementation of treatment strategies, there is a need to focus on screening, diagnosing those with HCV infection and population education strategies to break the chain of transmission.

With the success of the Punjab Model, where a cure rate of more than 90% was achieved among those enrolled,[15] and in concurrence with the global effort to eliminate HCV infection, India launched the National Viral Hepatitis Control Programme (NVHCP) in July 2018 which aims at improving access to diagnosis and treatment for viral hepatitis.[17] This programme covers the entire gamut of acute and chronic viral hepatitis including types A, B, C, D and E with a combined strategy of prevention, immunisation for HBV, screening, detection and timely treatment.[17]

In view of these recent developments to tackle the HCV burden, the Punjab state government plans to move forward with screening in a phased manner. Screening could be done among high-risk groups (HRGs) or a universal screening strategy could be one option as identification of HRGs, which include patients who are frequently recipients of blood products (eg, patients with thalassemia), PWID, patients on haemodialysis, organ transplant recipients and people living with HIV (PLHIV).[17] However, targeting of sensitive groups, such as PWID and PLHIV, can be associated with stigma which may be accompanied by other political as well as practical challenges.[17] Another viable strategy could be screening around the age of presentation of the infection which is around 40 years from the database of the 48 088 patients treated under MMPHCRF.[16]

To inform this policy decision on the implementation of a screening and treatment programme in the state of Punjab, we undertook this analysis to assess the incremental cost per quality-adjusted life year (QALY) for one-time universal screening followed by treatment of HCV-infected population as compared with no screening in Punjab. This study also aims to inform on which screening technology represents value for money and

would yield a better operational feasibility given the fiscal space for funding the HCV screening programme. Thus, a budget impact analysis (BIA) was performed in addition to estimating the financial resources required in terms of initiating a screening programme in the state.

## METHODS
### Overview of analysis
Screening services were assumed to be delivered as a facility-based intervention where active screening for HCV cases would be performed at 22 district hospitals in the state of Punjab, via existing resources, which will act as integrated testing as well as treatment sites for HCV. Since screening is to be initiated in the state, this assumption was made in consultation with experts from 'National Taskforce to Combat Viral Hepatitis'. We modelled the lifetime costs and outcomes for screening and treating HCV-infected patients in Punjab considering three different scenarios: no universal screening and treatment of those detected in 'routine care' setting; scenario I—screening with ELISA followed by confirmatory HCV-RNA quantification; and scenario II—screening with rapid diagnostic test (RDT) kit followed by confirmatory HCV-RNA quantification. A lifelong time horizon was considered so as to account for all costs and consequences of the intervention for chronic HCV. The outcomes are valued in terms of the number of HCV deaths, life years and QALYs. All the future costs and consequences, that is, early versus late detection of HCV, new HCV infections averted and HCV deaths averted, are discounted at a rate of 3%. This choice of discounting rate is based on standard international guidelines along with being consistent with other Indian economic evaluations.[18–21] The methodological principles are in concurrence with the Indian reference case for conducting economic evaluations published by the health technology agency in India.[21]

### No screening and treatment/do-nothing/routine-care scenario
The routine-care scenario represents the current situation where there is no screening programme. In this situation, we assume that patients present to the facility of their own volition when symptoms develop and access the required management support. Since the treatment coverage in India is low,[1 22] we also assume that only 5% of the HCV-infected cohort receive the testing and treatment on their own and the rest progress to advance stages of HCV as per the natural progression of the disease. The drug regimen followed to treat those infected is as per the national guidelines for diagnosis and management of HCV in India (figure 1). The rest 95% patients who become aware of their status at later stages with presentation of cirrhosis incur the cost of supportive management, including the management of associated complications like decompensated disease or HCC.

### Scenario I: screening with ELISA followed by confirmatory test and treatment
In this scenario, ELISA was used as the screening test followed by HCV-RNA as the confirmatory test. The

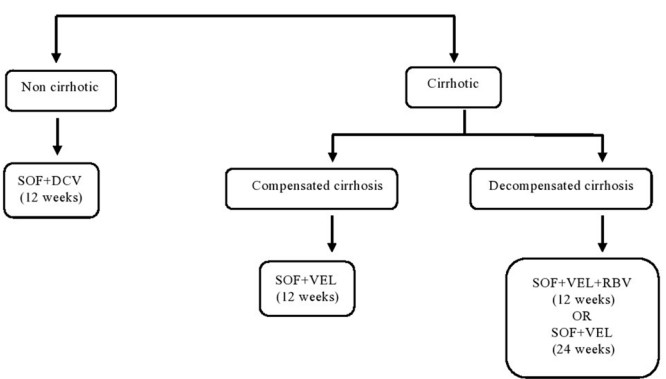

**Figure 1** Treatment guidelines for management of chronic hepatitis C virus infection.[60] DCV, daclatasvir; RBV, ribavirin; SOF, sofosbuvir; VEL, velpatasvir.

coverage of screening has been assumed to be 30% based on the uptake of screening in other programmes.[23–25] Those who are then determined eligible are administered treatment as per the algorithm. A loss to follow-up of 10% has been assumed at each step from testing to treatment. Currently, ELISA is being used at the facility level for testing antibodies to HCV. This scenario is compared with the do-nothing/routine-care scenario where there is no active screening programme in practice.

### Scenario II: screening with RDT followed by confirmatory test and treatment

This scenario differs from the above in the test used for screening. RDT kits are considered instead of ELISA for testing the antibodies to HCV followed by HCV-RNA as the confirmatory test and thus treatment. The rationale for comparing both these testing strategies is that use of ELISA-based screening tests is associated with long turn-around time, high cost and requirements for specialised apparatus and trained technicians, whereas screening with RDTs obviates the need for multiple follow-up appointments as well as shortens the waiting time. Also, RDTs follow a simpler technique as compared with ELISA-based screening.[26]

In both scenarios, all eligible viremic HCV-infected persons will receive free-of-charge DAAs as per the treatment algorithm under the NVHCP. The coverage and loss to follow-up assumptions remain the same for either scenario. Similar to above, this strategy is also compared with the routine-care scenario.

### Model structure

A decision tree (figure 2) integrated with a mathematical Markov model (figure 3), depicting natural history of HCV, was prepared in Microsoft Excel to simulate the progression of disease. The costs and health outcomes of each intervention scenario (I and II) are compared with the routine-care scenario.

A cross-sectional study in Punjab, which included 5543 individuals, reported a prevalence of 3.6% with a viraemia rate of 70%, thus estimating a burden of around 700 000 patients with viraemic hepatitis C.[13] This study also

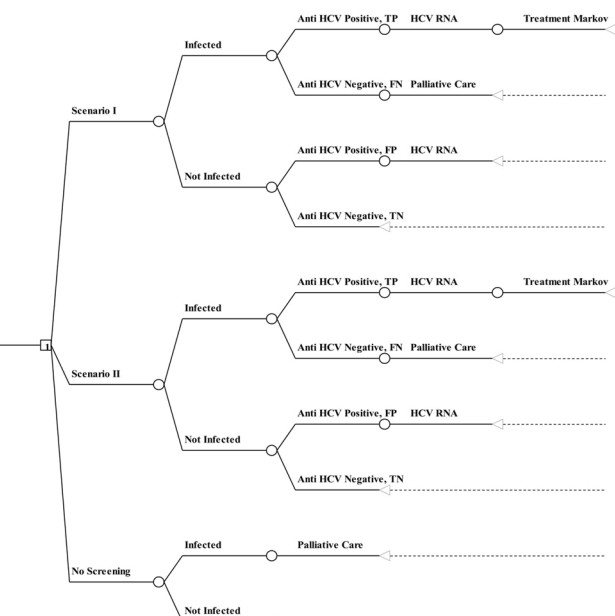

**Figure 2** Decision tree depicting the screening pathway for different scenarios. FN, False Negative; FP, False Positive; HCV, hepatitis C virus; TN, True Negative; TP, True Positive.

reports age-based prevalence of chronic HCV infection in Punjab.[13] Based on the analysis of the MMPHCRF patient data, the mean age of presentation of HCV infection was estimated to be around 40 years.[16] Thus, in our analysis, we considered one-time universal screening of the Punjab population around the mean age of presentation of HCV. Subsequent to treatment, the patients were divided into those who achieve sustained virologic response (SVR), a surrogate marker for virologic cure, and others who do not (treatment failures). The HCV-infected persons in fibrosis stages F0–F3, who achieve SVR, are not likely to progress further and thus the progression is halted.[27] Patients with cirrhosis who achieve SVR continue to progress to subsequent stages of liver disease progression, although at a slower rate as compared with the natural progression rates and with a lower risk of HCC.[27] Those

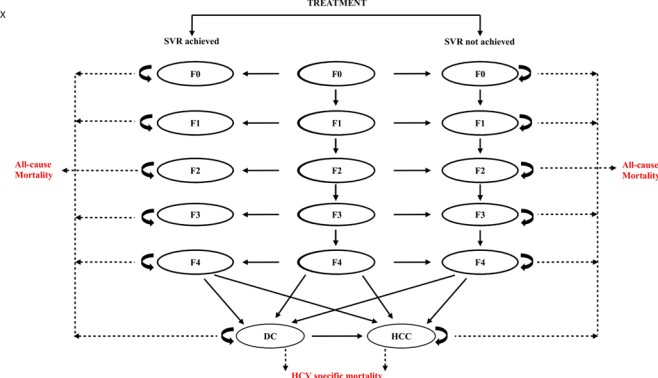

**Figure 3** Markov model depicting the natural history of hepatitis C infection. DC, decompensated cirrhosis; HCC, hepatocellular carcinoma; HCV, hepatitis C virus; SVR, sustained virologic response.

patients who do not achieve SVR continue to progress to subsequent stages as per the natural progression of the disease.[27] It was assumed that death due to HCV occurs from decompensated cirrhosis (DC) and HCC states while all-cause mortality happens in all stages. A cycle length of 1 year was considered appropriate along with a lifetime study horizon to comprehensively capture all the costs and benefits of the intervention.

## Costing

A societal perspective was considered to analyse total costs for screening and treatment. Apart from the cost of screening and confirmatory tests, treatment-associated costs were also included, that is, cost of outpatient (OPD) consultation for therapeutic purpose, inpatient (IPD) care for hospitalisation which is required in advanced stages as well as cost of management of complications that occur at each stage of the disease. The costs of specific procedures required in assessment of fibrosis and at later stages, such as endoscopy, transient elastography, ultrasound and ascitic tap, were also included. Average costs for primary, secondary and tertiary OPD consultation and hospitalisation as reported in recently published Indian studies, as well as National Health System Cost Database, were used (table 1).[28–30] The tertiary sector costs for IPD included both the cost of medicine ward[31] as well as the intensive care unit costs. The costs of ELISA and RDT kits were as per the procurement rates of the Punjab government obtained via consultations with programme officers. Other costs associated with the test such as cost of equipment, human resource, infrastructures, drugs and consumables, and other non-consumable items were available from data collected for costing the laboratory procedures at the district hospital. The confirmatory test HCV-RNA was outsourced by the government, thus the provider payment rates of Punjab government contracted laboratories were used.[16] The cost of DAAs including ribavirin (RBV), ranging from 4000 to 17000 Indian rupees, US$57–242 per patient, was also as per the procurement prices in Punjab.[15] All costs are also reported for the current year in US$ at a currency exchange rate of US$1=70.5 Indian rupees.[32]

We also included the out-of-pocket (OOP) expenditure of the patient for OPD and hospitalisation in public and private facilities. It was derived based on analysis of unit-level data from National Sample Survey 71st round, conducted during January–June 2014, which is a large household survey covering a sample of 65 932 households.[33]

## Valuation of consequences

The impact of intervention was dependent on the sensitivity and specificity of the screening tests, the compliance of each step as well as SVR rates achieved with the treatment provided subsequently. The estimates for sensitivity and specificity of ELISA and RDT fed into the model have been reported by published systematic reviews and meta-analysis.[34 35] It was assumed that 10% (5%–20%) of the

population was lost during follow-up at each step. The impact of treatment has been reported in terms of SVR which was modelled to assess progression to more severe health states and finally in terms of survival. The health-related quality of life was measured using Euro-QoL five dimensions questionnaire which was administered to a total of 230 patients who were being treated for chronic HCV infection.[27] We assumed that the quality of life among patients without cirrhosis who achieve SVR to be equivalent to the general population[36] ; whereas for the patients with cirrhosis and HCC, we assumed that the quality of life score was similar as in the respective advanced stages.

The estimates for SVR rates were obtained from MMPHCRF data of 48 808 patients.[16 27] All the parameters pertaining to progression of HCV, with and without achieving SVR, were obtained through published literature.[37 38] Annual mortality estimates from DC and HCC were 21.6% and 41.1%, respectively, and were reported from published literature.[39–41] The age-wise all-cause mortality was obtained from the Sample Registration System report for Punjab state.[42]

Also, we made an assumption regarding the coverage of screening to be 30% in the base case.[23–25] Since there is no screening programme currently running in the state, we tried to put in a modest estimate which is not very high as well as will help to reflect the impact of screening activity.

All the parameters related to costs and effectiveness have been reported in table 1.

## Sensitivity analysis

We undertook a probabilistic sensitivity analysis (PSA) to analyse the effect of joint parameter uncertainty on the incremental cost-effectiveness ratio (ICER).[43 44] For several parameters, 95% CIs were available such as age-wise prevalence estimates, progression parameters, sensitivity and specificity of screening tests. For other parameters including the stage-wise distribution at the time of diagnosis and SVR rates were varied by 5% for both upper and lower limits, RBV tolerance was varied by 10% for both upper and lower limits. All the drug costs were varied by 20% on the lower side only as the current prices of DAA are the negotiated prices resulting from operating bulk procurement mechanisms in the state. Therefore, the prices might decrease further but the increase in the prices is unlikely. The cost of screening tests was varied 20% on the lower side and 50% on the upper side. For quality of life estimates, SEs were computed from the primary analysis.

For undertaking the PSA, gamma distribution was used for all the cost parameters, a beta distribution was used in case of parameters where 95% CIs were available, and a uniform distribution was applied where the upper and lower limits were available. Monte Carlo method was used, and the results were simulated 1000 times. We computed the median estimates along with 2.5th and 97.5th percentile to estimate 95% CI.

Based on the PSA, we estimated the probability of being cost-effective at varying willingness to pay thresholds for

**Table 1** Demographic, epidemiological, effectiveness and cost-related parameters

| Parameters | Base value | Lower limit | Upper limit | Distribution | Source |
|---|---|---|---|---|---|
| **Epidemiological parameters** | | | | | |
| Viraemic prevalence of HCV in Punjab | 0.026 | 0.0234 | 0.0286 | Uniform | 12 13 15 |
| 19–29 years | 0.012 | 0.005 | 0.019 | Uniform | 12 13 15 |
| 30–39 years | 0.031 | 0.018 | 0.043 | Uniform | 12 13 15 |
| 40–49 years | 0.047 | 0.031 | 0.062 | Uniform | 12 13 15 |
| 50–59 years | 0.045 | 0.027 | 0.062 | Uniform | 12 13 15 |
| ≥60 years | 0.027 | 0.014 | 0.039 | Uniform | 12 13 15 |
| Stage-wise distribution of HCV at diagnosis | | | | | |
| F0 | 0.325 | 0.30875 | 0.34125 | Uniform | 16 27 |
| F1 | 0.325 | 0.30875 | 0.34125 | Uniform | 16 27 |
| F2 | 0.1 | 0.095 | 0.105 | Uniform | 16 27 |
| F3 | 0.1 | 0.095 | 0.105 | Uniform | 16 27 |
| F4 | 0.12 | 0.114 | 0.126 | Uniform | 16 27 |
| DC | 0.03 | 0.0285 | 0.0315 | Uniform | 16 27 |
| Proportion of population that is RBV tolerant | 0.9 | 0.81 | 0.99 | Uniform | 16 27 |
| **Clinical parameters** | | | | | Expert opinion |
| F0–F3 | | | | | |
| Number of OPD contacts required | 3 | 2 | 4 | Uniform | |
| Proportion of patients requiring hospitalisations | 0.05 | 0.01 | 0.1 | Uniform | |
| Number of hospitalisations per patient per year | 2 | 1 | 4 | Uniform | |
| F4 | | | | | |
| Number of OPD contacts required | 3 | 2 | 4 | Uniform | |
| Proportion of patients requiring hospitalisations | 0.05 | 0.01 | 0.1 | Uniform | |
| Number of hospitalisations per patient per year | 2 | 1 | 4 | Uniform | |
| DC | | | | | |
| Number of OPD contacts required | 12 | 6 | 18 | Uniform | |
| Proportion of patients requiring hospitalisations | 0.8 | 0.7 | 0.9 | Uniform | |
| Number of hospitalisations per patient per year | 6 | 4 | 8 | Uniform | |
| HCC | | | | | |
| Number of OPD contacts required | 12 | 6 | 18 | Uniform | |
| Proportion of patients requiring hospitalisations | 0.6 | 0.5 | 0.7 | Uniform | |
| Number of hospitalisations per patient per year | 2 | 1 | 4 | Uniform | |
| **Effectiveness parameters** | | | | | |
| Quality of life weights | | | | | |

**Table 1** Continued

| Parameters | Base value | Lower limit | Upper limit | Distribution | Source |
|---|---|---|---|---|---|
| F0–F3 | 0.63 | 0.57 | 0.70 | Beta | [27] |
| F4 | 0.56 | 0.51 | 0.61 | Beta | [27] |
| DC | 0.44 | 0.38 | 0.49 | Beta | [27] |
| HCC | 0.44 | 0.38 | 0.49 | Beta | [27] |
| F0–F3 (post-SVR) | 1 | 0.83 | 1 | Beta | [27 36] |
| Discount rate | 0.03 | 0.02 | 0.08 | Uniform | [18–21] |
| Transition probabilities | | | | | |
| F0 to F1 | 0.177 | 0.104 | 0.13 | Beta | [38 39] |
| F1 to F2 | 0.085 | 0.075 | 0.096 | Beta | [38 39] |
| F2 to F3 | 0.12 | 0.109 | 0.133 | Beta | [38 39] |
| F3 to F4 | 0.116 | 0.104 | 0.129 | Beta | [38 39] |
| F4 to DC | 0.035 | 0.027 | 0.043 | Beta | [38 39] |
| F4 to DC (post-SVR) | 0.002 | 0.0001 | 0.005 | Beta | [38 39] |
| F4 to HCC | 0.024 | 0.018 | 0.031 | Beta | [38 39] |
| F4 to HCC (post-SVR) | 0.005 | 0.001 | 0.009 | Beta | [38 39] |
| DC to HCC | 0.068 | 0.03 | 0.083 | Beta | [38 39] |
| DC to HCC (post-SVR) | 0.03 | 0.0225 | 0.0375 | Beta | [38 39] |
| Probability of dying due to HCV | | | | | |
| F0–F4 | 0 | 0 | 0 | Uniform | [27] |
| DC | 0.216 | 0.162 | 0.27 | Uniform | [40–42] |
| HCC | 0.411 | 0.31 | 0.51 | Uniform | [40–42] |
| SVR rates (%) | | | | | |
| SOF+DCV (12 weeks) | 74 | 66.5 | 73.5 | Uniform | [16 27] |
| SOF+VEL (12 weeks) | 84 | 79.8 | 88.2 | Uniform | [16 27] |
| SOF+VEL (24 weeks) | 81 | 76.95 | 85.05 | Uniform | [16 27] |
| SOF+VEL+RBV (12 weeks) | 84 | 79.8 | 88.2 | Uniform | [16 27] |
| Sensitivity ELISA | 0.952 | 0.918 | 0.972 | Beta | [34 35] |
| Specificity ELISA | 0.986 | 0.976 | 0.993 | Beta | [34 35] |
| Sensitivity RDT | 0.984 | 0.889 | 0.998 | Beta | [34 35] |
| Specificity RDT | 0.986 | 0.949 | 0.996 | Beta | [34 35] |
| Coverage parameters | | | | | |
| Treatment coverage in do-nothing scenario (%) | 5 | 0 | 10 | Uniform | Author's assumption |
| Coverage of screening in scenario I and II (%) | 30 | 20 | 50 | Uniform | Author's assumption |
| Loss to follow-up at each step (%) | 10 | 5 | 20 | Uniform | Author's assumption |
| **Cost parameters** | | | | | |
| Drug costs—12 weeks (Indian rupees) | | | | | |
| SOF+DCV | 4509 | 3607.2 | 4509 | Gamma | [16] |
| SOF+VEL | 13104 | 10483.2 | 13104 | Gamma | [16] |
| SOF+VEL+RBV | 16818 | 13454.4 | 16818 | Gamma | [16] |
| Cost of diagnostic tests (Indian rupees) | | | | | |
| ELISA | 72.95 | 58.36 | 109.425 | Gamma | [16] |
| RDT | 39 | 31.2 | 58.5 | | [16] |

Continued

**Table 1** Continued

| Parameters | Base value | Lower limit | Upper limit | Distribution | Source |
|---|---|---|---|---|---|
| HCV-RNA | 880 | 704 | 1320 | Gamma | [16] |
| Routine tests (CBC, LFT, creatinine) | 500 | 400 | 750 | Gamma | [16] |
| Cirrhosis evaluation (FibroScan) | 350 | 280 | 525 | Gamma | [16] |
| Genotyping | 895 | 716 | 1342.5 | Gamma | [16] |
| Cost per OPD consultation (Indian rupees) | | | | | |
| Primary | 1686.3 | 1180.41 | 2192.19 | Gamma | [28–31] |
| Secondary | 1734 | 1213.8 | 2254.2 | Gamma | [28–31] |
| Tertiary | 2024 | 1416.8 | 2631.2 | Gamma | [28–31] |
| Cost per patient hospitalisation (Indian rupees) | | | | | |
| Primary | 6347.1 | 4442.97 | 8251.23 | Gamma | [28–31] |
| Secondary | 7597 | 5317.9 | 9876.1 | Gamma | [28–31] |
| Tertiary | 18693 | 13085.1 | 24300.9 | Gamma | [28–31] |
| Per unit cost of integrated training (Indian rupees) | | | | | |
| District | 50000 | 40000 | 75000 | Gamma | Expert opinion |
| State | 308000 | 246400 | 462000 | Gamma | Expert opinion |
| Per district cost of IEC/BCC (Indian rupees) | 4000 | 3200 | 6000 | Gamma | Expert opinion |

CBC, complete blood count; DC, decompensated cirrhosis; DCV, daclatasvir; HCC, hepatocellular carcinoma; HCV, hepatitis C virus; IEC/BCC, information education counselling/behaviour change communication; LDV, Ledipasvir; LFT, liver function test; OPD, outpatient department; RBV, ribavirin; RDT, rapid diagnostic test; SOF, sofosbuvir; SVR, sustained virologic response; VEL, velpatasvir.

scenarios I and II compared with no screening. We used a one-time per-capita Gross Domestic Product (GDP) value for India as the threshold to base our conclusion about cost-effectiveness. This was in view of the national guidelines for Health Technology Assessment (HTA),[21] current practice in Indian economic evaluations[20 45] and more national discussions.[46]

A scenario analysis was conducted for assessing the cost-effectiveness of scenarios I and II as compared with the routine care, where we modelled the impact of screening and treatment for all people aged 18 years and above. Thus, this scenario analysis primarily evaluated the cost-effectiveness of screening which could be attributed to difference in age of initiating the screening.

We also conducted a univariate analysis to assess the impact of varying screening coverage parameter. Thus, we estimated ICERs at coverage rates of 10%, 20%, 50%, 70% and 100%.

### Budget impact analysis

We undertook a BIA to assess the requirement of resources for the uptake of screening and treatment programme. Two scenarios were considered, screening everyone aged 18 years and above and screening the age cohort of 40–45 years which is the mean age of presentation of HCV infection. The parameters considered were: Punjab population estimates from census,[47] prevalence estimates from published literature,[12 13 15] cost of screening tests

which included ELISA and RDT based on expert consultations with programme officers, cost of confirmatory test HCV-RNA,[16] cost of drugs given for treatment as per the provider payment rates of Punjab government,[16] cost incurred on human resources in terms of their salary as well as cost of integrated trainings conducted to train personnel for carrying out screening activities, and the cost spent on information education counselling (IEC) and behaviour change communication (BCC) activities. The costs of integrated trainings and IEC/BCC activities were derived from expert consultations with programme officers and the review of state budget documents.

For the human resources, we estimated the man-hours currently being dedicated to HCV screening, therefore we budgeted in terms of additional man-hours required to screen a given proportion of population if screening facilities were to be delivered via existing set-up. Costs of training and IEC/BCC have been treated as fixed costs as there is no basis for determining repeated IEC/BCC activities in absence of an active screening programme. The costs of screening test, confirmatory test and drugs are the variable costs associated with screening and treatment.

A payer's perspective was considered for the analysis with no discounting and a time horizon of 5 years. Results were reported at different coverage rates of screening 10%–100%. The methodology of the BIA was

in consistency with the International Society for Pharmacoeconomics and Outcomes Research reporting framework.[48]

## Patient and public involvement

No patient and public involved.

## RESULTS
### Costs

In the routine scenario, a lifetime cost of 2775 Indian rupees (US$38.8) is incurred per person. In scenario I, a lifetime societal cost of 998 Indian rupees (US$14.1) is incurred per person; whereas in scenario II, 947 Indian rupees (US$13.2) per person are incurred. When scenario I is compared with no screening option, it saves 1810 Indian rupees (US$25.3) per person. The cost savings are similar when scenario II is compared with no screening, that is, 1812 Indian rupees (US$25.3) per person.

### Valuation of consequences

The life years lived by Punjab population including HCV-infected cohort as well as in no screening, scenarios I and II were 15.75, 15.92 and 15.76 per person, respectively. Similarly, the QALYs lived per person in no screening, scenarios I and II were 15.57, 15.78 and 15.6, respectively. As compared with the routine scenario, there was a gain of 0.0046 (0.0013 to 0.011) life years and 0.028 (0.008 to 0.06) QALY per person in scenario I. The gains are similar when scenario II is compared with no screening, that is, 0.0045 (0.0012 to 0.012) life years and 0.027 (0.008 to 0.061) QALY per person.

Costs and outcomes for each scenario have been given in table 2.

### Cost-effectiveness

When compared with no screening, both scenarios I and II dominate over no screening, that is, they have better health outcomes along with being cost-saving. However, when the incremental gains of scenarios I and II (relative to no screening) are compared, there was an overlap between the CIs of the incremental gains for both costs and outcomes which implies that the difference is statistically insignificant.

In addition, when compared with scenario I, scenario II results in an incremental gain of 0.0003 life years and 0.001 QALYs per patient with a saving of 13.73 Indian rupees per patient. The reason for minimal gains is that sensitivity and specificity of both ELISA and RDT are overlapping. However, the cost of RDT is less than ELISA.

Screening is cost-saving at all coverage levels from 10% to 100%. Scenarios I and II (relative to no screening) have a 100% probability of being cost-saving. Also, screening with both, either ELISA or RDT, followed by treatment for people aged 18 years and above, is cost-saving. This implies that irrespective of the age of initiation of screening and subsequent treatment, screening is a dominant strategy as compared with no screening. Figure 4

**Table 2** Cost and effects of screening and treating chronic hepatitis C virus-infected patients

| | |
|---|---|
| **No screening** | |
| Life years per patient | 15.75 (11.79 to 23.85) |
| QALY per patient | 15.57 (11.63 to 23.60) |
| Cost per patient (Indian rupees) | 2775 (1549 to 4898) |
| **Scenario I** | |
| Life years per patient | 15.92 (11.79 to 23.86) |
| QALY per patient | 15.78 (11.65 to 23.67) |
| Cost per patient (Indian rupees) | 998 (456 to 1932) |
| **Scenario II** | |
| Life years per patient | 15.75 (11.79 to 23.86) |
| QALY per patient | 15.60 (11.65 to 23.63) |
| Cost per patient (Indian rupees) | 947 (468 to 1837) |
| **Incremental gains** | |
| Scenario I–no screening | |
| Life years gained | 0.0046 (0.0013 to 0.011) |
| QALYs gained | 0.028 (0.008 to 0.061) |
| Cost difference (Indian rupees) | −1810 (−3376 to −867) |
| Scenario II–no screening | |
| Life years gained | 0.0045 (0.001 to 0.012) |
| QALYs gained | 0.027 (0.008 to 0.061) |
| Cost difference (Indian rupees) | −1812 (−3468 to −850) |

QALY, quality-adjusted life year.;

displays cost-effectiveness planes for both the scenarios (ELISA and RDT) as compared with no screening.

### Budget impact analysis

Screening the population of 18 years and above, at base coverage of 30%, ELISA costs 1779 million Indian rupees, that is, 3.8% of the Punjab state health budget[49]; whereas with RDT 1586 million Indian rupees, that is, 3.4% of the state health budget. However, screening the age cohort of 40–45 years, with ELISA and RDT, accounts for 0.45% and 0.41% of the state health budget, respectively.

In addition, given the current human resources, who perform other tests in addition to HCV, screening those 18 years and older at a coverage of 30%, would require an additional 186 336 man-hours (1260 persons). Similarly, screening the age cohort of 40–45 years would require an additional 14 665 man-hours (99 persons). However, if one dedicated person to HCV screening is deployed at each of the districts, the human resources will be sufficient to screen the age cohort of 40–45 years; whereas, additional 1015 persons would be required to screen everyone 18 years and older.

The year-wise disaggregated results from the first to fifth year (table 3A), distribution of budget estimates for the base year at 30% coverage according to type of resources (table 3B) and budget estimates for different coverage levels (table 3C) are summarised under tables 3A–C.

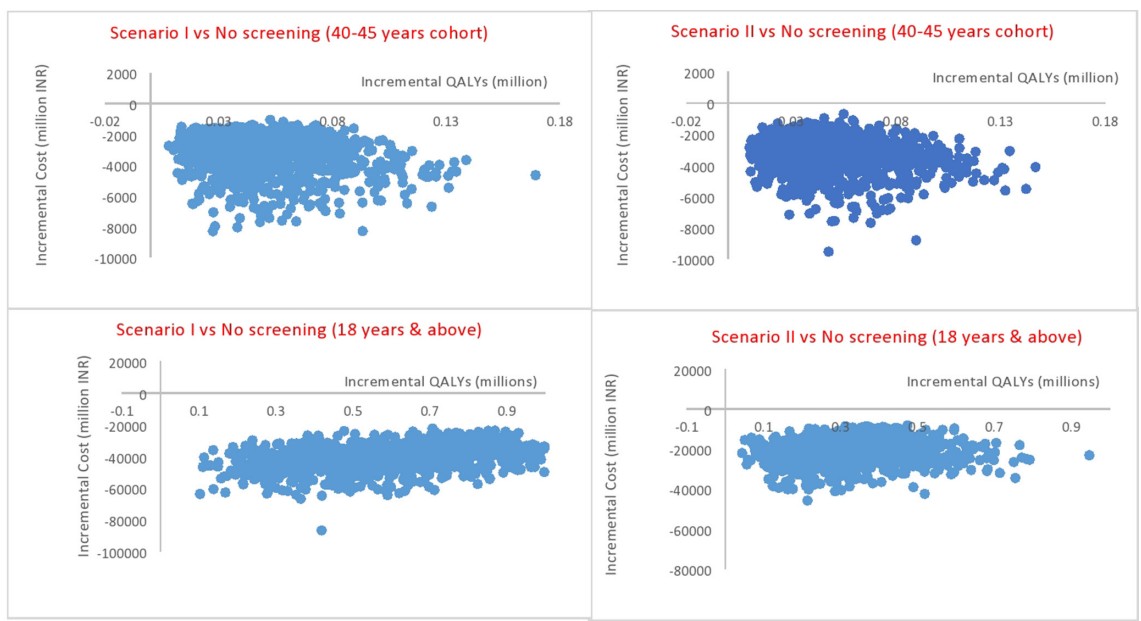

**Figure 4** Cost-effectiveness planes for scenarios I and II versus no screening. INR, Indian rupee; QALY, quality-adjusted life year.

## DISCUSSION

In resource-limited settings like India, where OOP expenditure accounts for 62% of the total health expenditure,[50] it is of utmost importance to use the resources in a sustainable manner. The present analysis assessed the sustainability of a proposed screening and treatment programme for HCV in the setting of the existing NVHCP. Our findings demonstrate that screening, either with ELISA or RDT, is dominant over no screening which implies having higher QALYs gained and lower costs. Also, the difference between incremental gains for both scenarios I and II (compared with no screening) is insignificant which can be attributed to overlapping sensitivity and specificity of ELISA and RDT.

Treating HCV infection in early stages results in halting the progression to advanced stages which are associated with high management costs.[28] Our previous analysis demonstrated that, given a lifetime horizon, treatment with DAA decreases the cases of DC and HCC attributable to HCV by half as compared with no treatment.[27] Second, the effectiveness rates of the DAA-based treatment are as high as 90%, even in public health programmes.[51–53] Third, the prices of DAA and diagnostic tests have been significantly reduced through price negotiation with bulk procurement of the efficacious generic drugs and reagents. All these factors support the cost-saving nature of the screening and treatment intervention for HCV in Punjab.

Though the intervention of screening appears to be dominant, it is associated with a huge budget impact. BIA are financial tools which help us to estimate the future spending required in the chosen time horizon.[54] In light of the results of BIA, one-time screening of population aged 18 years and older would cost in the range of 3.4% (RDT, scenario II) to 3.8% (ELISA, scenario I) of the state health budget;

**Table 3A** Budget Impact Assessment: Incremental Budgetary estimates spread across the time horizon (year 1 to year 5) in million rupees (million US $)

|  |  | ELISA | RDT |
|---|---|---|---|
| 18 years and older | Year 1 | 1779.42 (25.24) | 1586.32 (22.5) |
|  | Year 2 | 52.81 (0.75) | 47.1 (0.67) |
|  | Year 3 | 53.05 (0.75) | 47.54 (0.67) |
|  | Year 4 | 53.44 (0.76) | 48.27 (0.68) |
|  | Year 5 | 53.93 (0.76) | 49.16 (0.7) |
| 40–45 years | Year 1 | 208.7 (2.96) | 190.46 (2.7) |
|  | Year 2 | 45.83 (0.65) | 41.83 (0.59) |
|  | Year 3 | 46.07 (0.65) | 42.07 (0.6) |
|  | Year 4 | 46.46 (0.66) | 42.46 (0.6) |

RDT, rapid diagnostic test.

**Table 3B** Budget Impact Assessment: Resource wise disaggregated estimates of budget for the 1st year at base coverage (30%) in million rupees (million US $)

| | 40–45 years | | 18 years and older | |
| --- | --- | --- | --- | --- |
| | **ELISA** | **RDT** | **ELISA** | **RDT** |
| HR | 1.314 (0.019) | 1.314 (0.019) | 3.367 (0.048) | 3.367 (0.048) |
| IEC/BCC | 0.088 (0.001) | 0.088 (0.001) | 0.088 (0.001) | 0.088 (0.001) |
| Screening Test | 35.94 (0.51) | 17.7 (0.251) | 380.39 (5.396) | 187.29 (2.657) |
| Confirmatory Test | 22.12 (0.314) | 22.12 (0.314) | 180.16 (2.555) | 180.16 (2.555) |
| Drugs | 149.24 (2.117) | 149.24 (2.117) | 1215.42 (17.24) | 1215.42 (17.24) |
| Total | 208.702 (2.96) | 190.462 (2.702) | 1779.425 (25.24) | 1586.325 (22.501) |

HR, Human resources; IEC/BCC, Information-Education-Counselling-Behaviour-Change-Communication; RDT, Rapid Diagnostic Tests.

whereas, screening the population in the age group of 40–45 years old will require spending 0.41% (RDT, scenario II) to 0.45% (ELISA, scenario I) of the state health budget. One way to assess whether the additional budget for health will be able to accommodate for higher allocation to any programme is the analysis of the fiscal space. Fiscal space is dependent on[55]: the macroeconomic factors for which the impact on the GDP is assessed. Second, one can generate the estimates for fiscal space through budget reprioritisation. Third, we can measure efficiency and estimate monetary value of efficiency enhancing measures. Consequent to this, we compare the required budget for screening with average annual growth in the health budget of Punjab state, which has been reported to 10%.[56] According to the India State-Level Disease Burden Initiative, cirrhosis and other chronic liver diseases account for 1.44% of total morbidity burden in Punjab.[57] Given this scenario, spending more than 0.144% of the health budget for HCV programme seems unrealistic. Thus, opting for screening the age group of 40–45 years with RDT instead of all aged 18 years and older appears operationally feasible.

Our findings are consistent with other studies in the same domain.[58–60] A model-based analysis evaluated the cost-effectiveness of one-time screening of individuals between 40 and 70 years of age compared with no screening in Korea. The findings of the study demonstrate that one-time screening for HCV was highly cost-effective in comparison with their routine scenario, that is, no screening.[58] Another model-based cost-effectiveness analysis for screening and treatment of HCV in Egypt also demonstrates one-time screening and treatment to be a dominant strategy as compared with no screening.[59]

Further, a cost-effectiveness study based in Japan also evaluates the cost-effectiveness of screening and treatment. They compare no screening versus screening and interferon-based treatment versus screening and DAA-based treatment. Also, they stratified the analysis according to different age groups. Similar to our results, this study also concluded that screening the age group of 40–49 years followed by DAA-based treatment is the most cost-effective strategy.[60] Another study based in the USA compared one-time universal screening as well as birth cohort screening with risk factor screening. Both types of interventions were deemed cost-effective as compared with risk factor-based screening, whereas screening among the general population remained cost-effective under all assumptions.[61]

Studies conducted pertaining to assess the cost-effectiveness of screening in LMICs are very few. Thus, our analysis would contribute to generate evidence pertaining to this area. We used a biologically plausible model which is fitted with local data on disease epidemiology and effectiveness of DAAs which is drawn from the

**Table 3C** Budget Impact Assessment: Estimates for different screening scenarios in million rupees (million US $)

| | 18 years and older | | Age cohort 40-45 years | |
| --- | --- | --- | --- | --- |
| **Coverage of screening (%)** | **ELISA** | **RDT** | **ELISA** | **RDT** |
| 10 | 593.14 (8.41) | 528.78 (7.5) | 69.57 (0.99) | 63.49 (0.9) |
| 20 | 1186.28 (16.83) | 1057.54 (15) | 139.13 (1.97) | 126.97 (1.8) |
| 30 | 1779.42 (25.24) | 1586.32 (22.5) | 208.7 (2.96) | 190.46 (2.7) |
| 50 | 2965.69 (42.07) | 2643.87 (37.5) | 347.84 (4.93) | 317.44 (4.5) |
| 70 | 4151.97 (58.89) | 3701.41 (52.5) | 486.98 (6.91) | 444.42 (6.3) |
| 100 | 5931.39 (84.13) | 5287.73 (75) | 695.68 (9.87) | 634.88 (9.01) |

RDT, Rapid Diagnostic Tests.

programme data of 48 808 patients with HCV on treatment in Punjab. Second, cost of treatment is drawn from the National Health System Cost Database and nationally representative household surveys for patient costs. Similarly, the cost of screening is also derived from the real-world application in local context. Third, the data on quality of life are also based on our analysis of primary data collected locally. Thus, all the parameters used in the model are estimates from the real-world setting for which the analysis is being conducted.

We do acknowledge certain limitations of the present study. We did not consider the possibility of reinfection once treated in our model as we were estimating the impact of one-time screening and treatment. Therefore, we assumed that including reinfection would not change the direction of results. Second, the sample sizes for estimating the utility values may not be large enough for different health states. However, this is likely to influence the SEs; the effect of which is likely to be captured in our PSA.

We would also like to mention that we did not include testing for HBV and HIV among those who tested positive for HCV. Fourth, we assumed the same rate of lost to follow-up and adherence to treatment among patients diagnosed through universal screening and those who are seeking for treatment. However, these estimates are unlikely to alter the direction of results as the proportion of people seeking diagnostic and treatment services for HCV on their own is very low.[4 8 9]

Finally, we did not include productivity losses in our cost analysis. However, as it has also been argued that it is somewhat reflected in the valuation of consequences when trade-off techniques are used to elicit quality of life.[62] Moreover, there are methodological as well as parameter uncertainties which are likely to be introduced when productivity losses are accounted for. Finally, the national HTA guidelines for India have not recommended inclusion of productivity losses.

## CONCLUSION

Our findings suggest that one-time universal screening followed by treatment of HCV infection is a dominant strategy as compared with no screening. Though screening is deemed cost-saving at all coverage rates, the budget impact of screening programme is very high. Thus, based on the ground of both cost-effectiveness and budget impact we recommend screening the age cohort around the mean age of presentation of disease instead of the whole population with RDT instead of ELISA. Implementation of a screening programme can improve the uptake of available low-cost and highly efficacious drug treatment. Reforms should be introduced to improve both screening and treatment activities to cope up with the rising burden of HCV.

**Author affiliations**
[1]Department of Community Medicine and School of Public Health, Post Graduate Institute of Medical Education and Research School of Public Health, Chandigarh, Punjab, India
[2]Department of Hepatology, Post Graduate Institute of Medical Education and Research, Chandigarh, India
[3]Department of Health and Family Welfare, National Viral Hepatitis Control Program, Government of Punjab, Chandigarh, India
[4]Director, Sanjay Gandhi Post Graduate Institute of Medical Sciences, Lucknow, Uttar Pradesh, India
[5]Health Intervention and Technology Assessment Program, Ministry of Public Health, Mueang Nonthaburi, Nonthaburi, Thailand
[6]Saw Swee Hock School of Public Health (SSHSPH), National University of Singapore, Singapore
[7]Department of Community Medicine and School of Public Health, Post Graduate Institute of Medical Education and Research, Chandigarh, India

**Contributors** Conception—SP, YC and RKD. Design of the work— YC, SP, YT and RKD. Acquisition—YC, MP, SP, RKD and GSG. Analysis—YC, MP, SP and YT. Interpretation of data—SP, YT, YC and RKD. Drafting the work—YC and MP. Revising it critically for important intellectual content—SP, YT, RKD and GSG. Final approval of the version to be published—YC, SP, YT, MP, RKD and GSG. Agreement to be accountable for all aspects of the work in ensuring that questions related to the accuracy or integrity of any part of the work are appropriately investigated and resolved—YC, SP, YT, MP, RKD and GSG.

**Funding** The Health Intervention and Technology Assessment Program (HITAP) provided technical supervision for this study under the auspices of the International Decision Support Initiative (iDSI). HITAP is funded by the Thailand Research Fund (TRF) under a grant for a Senior Research Scholar (RTA5980011). HITAP is supported by the International Decision Support Initiative (iDSI) to provide technical assistance on health intervention and technology assessment to governments in low-income and middle-income countries. iDSI is funded by the Bill and Melinda Gates Foundation (OPP1134345), the UK's Department for International Development and the Rockefeller Foundation.

**Disclaimer** The findings, interpretations and conclusions expressed in this article do not necessarily reflect the views of the funding agencies.

**Competing interests** None declared.

**Patient consent for publication** Not required.

**Ethics approval** The present study is a model-based analysis which did not involve any human subjects as well as primary data collection. Therefore, ethical approval was not required. Administrative approval was sought from the Department of Health and Family Welfare, Punjab, India.

**Provenance and peer review** Not commissioned; externally peer reviewed.

**Data availability statement** All data relevant to the study are included in the article.

**ORCID iDs**
Madhumita Premkumar http://orcid.org/0000-0003-2961-4148
Shankar Prinja http://orcid.org/0000-0001-7719-6986

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
