## [Reviewer comments · BMJ Open]

ARTICLE DETAILS

TITLE (PROVISIONAL)	Cost-effectiveness and budget impact analysis of facility-based screening and treatment of Hepatitis C in Punjab state of India.
AUTHORS	Chugh, Yashika; Premkumar, Madhumita; Grover, Gagandeep; Dhiman, Radha K.; Teerawattananon, Yot; Prinja, Shankar

VERSION 1 – REVIEW

REVIEWER	Deuffic-Burban Sylvie Inserm, France I report having received consultancy honoraria from Intercept.
REVIEW RETURNED	02-Sep-2020

GENERAL COMMENTS	General comments In Punjab, HCV prevalence is as high as 3.6%, but only 10% of HCV-infected individuals have been treated due to lack of awareness of disease status. A global effort has been initiated to reach the WHO objectives with the launch of a national program to fight against viral hepatitis, by improving access to diagnosis and treatment for viral hepatitis. In this context, being able to inform public health authorities of the cost-effectiveness of one-time universal screening followed by treatment of HCV-infected population compared to no screening in Punjab, as well as the 5-year sustainability of such intervention, is relevant. However, I have one concern. Given the budget impact associated with the one-time universal screening strategy, the authors compared the budget impact of age cohort screening. But why has this intervention not been evaluated in the cost-effectiveness analysis compared to no screening? It seems essential to me. There is no point in verifying the sustainability of an intervention whose effectiveness and cost-effectiveness have not been first demonstrated. I have also specific comments. Specific comments a/ Major 1. Methods: a. Scenarios: It is clearly specified for scenario of routine care that screening is followed by treatment (in a certain proportion) but not for scenarios 1 and 2 (abstract and main text page 8, lines 182-185). It must be added. b. Data on health-related utilities (Table 1): The magnitude of utility estimates obtained from primary data collected locally is smaller than those typically used in cost-effectiveness analyses. The reason may be that these utilities were estimated in treated patients; they therefore do not reflect the utilities linked to natural history. Thus, analysis with such data will favour screening and treatment, against no screening (non-conservative analysis).
---

	c. Costs (Table 1): It is required to present the costs by state of health, otherwise explain in detail the method applied for the implementation of these costs: how many consultations for treatment? how many hospitalizations at advanced stages? how many fibrosis assessments? ... As presented, it is not clear how the costs were integrated into the progression of the disease 2. Results: a. Regarding the cost-effectiveness analysis, first, as some methodological elements are missing, it is difficult to judge the relevance of the results of the analyses, in particular that scenarios 1 and 2 are cost-saving. Second, I understand the interest of presenting the results of scenarios 1 and 2 compared to routine care, but the cost-effectiveness / cost-utility analysis must be presented in an incremental way, i.e. from the cheapest to the most expensive. b. Regarding the budget impact analysis, we would have liked to compare results of scenario 1 and 2 against routine care. b/ Minor Abstract: some context for this study would be useful; also the methods are not sufficiently described (budgetary impact over what time horizon? Which scenario(s)?)
--	---

REVIEWER	Davoud Pourmarzi The Australian National University, Australia
REVIEW RETURNED	09-Sep-2020

GENERAL COMMENTS	Thank you for the opportunity to review this manuscript. Chugh et al. have tried to address one of the important questions that we are facing in regard to eliminating HCV. The analysis can help policymakers in Punjab to understand the cost-effectiveness of providing screening and treatment for HCV. However, there are many questions that need to be answered and areas of the manuscript need to be improved to provide sufficient information for policymakers. I strongly recommend this manuscript be reviewed by a specialist in health economy before making any decision. The manuscript needs to be edited for language. I have provided detailed comments in the attached document. - The reviewer provided a marked copy with additional comments. Please contact the publisher for full details.
--

VERSION 1 – AUTHOR RESPONSE

Reviewer(s)' Comments to Author:

Reviewer#1

General comments

1. In Punjab, HCV prevalence is as high as 3.6%, but only 10% of HCV-infected individuals have been treated due to lack of awareness of disease status. A global effort has been initiated to reach the WHO objectives with the launch of a national program to fight against viral hepatitis, by improving access to diagnosis and treatment for viral hepatitis. In this context, being able to inform public health authorities of the cost-effectiveness of one-time universal screening followed by treatment of HCV-infected population compared to no screening in Punjab, as well as the 5-year sustainability of such intervention, is relevant. However, I have one concern. Given the budget impact associated with the one-time universal screening strategy, the authors compared the budget impact of age cohort screening. But why has this intervention not been evaluated in the cost-effectiveness analysis

compared to no screening? It seems essential to me. There is no point in verifying the sustainability of an intervention whose effectiveness and cost-effectiveness have not been first demonstrated.

Response:

We thank you for highlighting this issue. We would like to clarify that both cost-effectiveness and budget impact has been evaluated for both the scenarios: screening 18 years or older as well as screening around the age-group of 40-45 years (mean age of HCV presentation). For both budget impact and cost-effectiveness analysis, the results of screening the age group of 40-45 years were presented as the base case, however, the screening of 18 years or older strategy was assessed as a part of the sensitivity analysis.

Since, both strategies, i.e, screening 18 years or older as well as screening the age group of 40-45 years, were deemed to be cost-saving, the budget impact assessment was the prime driver of the recommendations of the analysis which favour screening the age group of 40-45 years instead of all aged 18 years or above.

The details for the same are mentioned in the manuscript at Page no. 13, line no. 340-343; Page no. 15, line no.392-397; Figure 3.

Specific comments

1. Methods:

a. Scenarios: It is clearly specified for scenario of routine care that screening is followed by treatment (in a certain proportion) but not for scenarios 1 and 2 (abstract and main text page 8, lines 182-185). It must be added.

Response:

In view of the reviewer's comment, we have made changes to specify both screening and treatment in description of scenario 1 and 2. Page no. 3, line no. 96-100; Page no. 8, line no. 224-227, 232-234.

b. Data on health-related utilities (Table 1): The magnitude of utility estimates obtained from primary data collected locally is smaller than those typically used in cost-effectiveness analyses. The reason may be that these utilities were estimated in treated patients; they therefore do not reflect the utilities linked to natural history. Thus, analysis with such data will favour screening and treatment, against no screening (non-conservative analysis).

Response:

We thank the reviewer for making this observation.

Patient level data collection for estimating the health-related quality of life (HRQoL) was undertaken among those visiting the outpatient (OPD) and inpatient department (IPD) of a large public sector hospital. We included a total of 230 patients from those visiting the OPD and admitted in the IPD. The number of patients included in the present study is similar to other cost-effectiveness studies commissioned by the India Health Technology Assessment (HTA) agency. A few of these studies have been cited below:

- Chauhan AS, Prinja S, Srinivasan R, Rai B, Malliga JS, Jyani G, Gupta N, Ghoshal S. Cost effectiveness of strategies for cervical cancer prevention in India. *PLoS One*. 2020 Sep 1;15(9):e0238291.
- Jyani G, Chauhan AS, Rai B, Ghoshal S, Srinivasan R, Prinja S. Health-related quality of life among cervical cancer patients in India. *Int J Gynecol Cancer*. 2020 Aug 11:ijgc-2020-001455. doi: 10.1136/ijgc-2020-001455. Epub ahead of print. PMID: 32788265.
- Chauhan AS, Prinja S, Ghoshal S, Verma R. Cost-effectiveness of treating head and neck cancer using intensity-modulated radiation therapy: implications for cancer control program in India. *International Journal of Technology Assessment in Health Care*.:1-8.

We do acknowledge that the sample size may not be large enough for individual health states.

However, this is likely to influence the standard errors; the effect of which is likely to be captured in the probabilistic sensitivity analysis (PSA). Since our sensitivity analysis results are robust to variation

in parameter estimates, we feel that it is unlikely to affect the overall conclusion of the study. Nonetheless, we have added this in the limitation section of our manuscript.

Secondly, we believe that inclusion of patients from a health facility which are likely to be those who are under treatment is unlikely to favour the estimated of cost-effectiveness towards screening. This is because the effect of screening and treatment is likely to be manifested more as a result of a slower rate of progression of HCV patients to worse off health states. In view of this, patients recruited in the sample were classified into individual health states using standard clinical criteria and hence similar QoL values were used for both intervention and counterfactual scenarios. To reiterate, it is the differential rate of progression to worse off health states in the alternative scenarios that is likely to reflect the effect of screening and treatment.

c. Costs (Table 1): It is required to present the costs by state of health, otherwise explain in detail the method applied for the implementation of these costs: how many consultations for treatment? how many hospitalizations at advanced stages? how many fibrosis assessments? ... As presented, it is not clear how the costs were integrated into the progression of the disease

Response:

In light of reviewer comments, we have added the required details in the Parameter table under the heading 'clinical parameters' (Table 1).

2. Results:

a. Regarding the cost-effectiveness analysis, first, as some methodological elements are missing, it is difficult to judge the relevance of the results of the analyses, in particular that scenarios 1 and 2 are cost-saving. Second, I understand the interest of presenting the results of scenarios 1 and 2 compared to routine care, but the cost-effectiveness / cost-utility analysis must be presented in an incremental way, i.e. from the cheapest to the most expensive.

Response:

We thank the reviewer for seeking clarification over this. As mentioned in the previous comments, we have provided more details on the input parameters for the model which will help clarifying the results of the cost-effectiveness. Secondly, both the scenarios being cost-saving when compared to the routine care implies that the incremental gains are more as compared to the costs incurred for screening and treatment. This can be attributed to the additional costs incurred, due to late diagnosis and in the absence of timely treatment, for managing the complications of advanced stages (HCC and cirrhosis) which are very high. Thirdly, we agree that the results of cost-effectiveness analysis should be presented in an incremental way. Although, we would like to state that, when interventions are being delivered through a public health set-up, the government would either chose option I or II whichever has better outcomes and lower costs. Therefore, we presented the results for both scenarios in comparison to the routine care.

However, as suggested by the reviewer, when compared to scenario I (ELISA), scenario II results in an incremental gain (per patient) of 0.0003 life years and 0.001 QALYs with a saving of Rs 13.73 per patient.

The reason for minimal gains is that sensitivity and specificity of both ELISA and RDT are overlapping. However, the cost of RDT is lesser than ELISA.

b. Regarding the budget impact analysis, we would have liked to compare results of scenario 1 and 2 against routine care.

Response:

We thank the reviewer for seeking clarification on this.

We would like to clarify that in our results of budget impact analysis, we provide the incremental value of the budget relative to the routine care, i.e, no screening scenario. This represents the additionality in the budgetary requirement which is relative to the existing situation.

In order to compute this, we estimated the number of people screened every year in a time horizon of 5 years and further, we estimated the additional number and associated costs for those put on

treatment during this time horizon from the cost-effectiveness model. To reiterate, the presented values of BIA in table 3 in the manuscript are the incremental value of budgetary requirement relative to the routine care, wherein, there is no existing screening program, thus patients present to the health facility as and when required.

Also, we have made changes in the title of Table 3 so as to make this point explicit.

3. Abstract: some context for this study would be useful; also the methods are not sufficiently described (budgetary impact over what time horizon? Which scenario(s)?)

Response:

As recommended by the reviewer, we have elaborated the abstract to include the specific details. Changes made on Page no. 3, Line no. 84-85, 90-91.

Reviewer#2

General comments

Thank you for the opportunity to review this manuscript. Chugh et al. have tried to address one of the important questions that we are facing in regard to eliminating HCV. The analysis can help policymakers in Punjab to understand the cost-effectiveness of providing screening and treatment for HCV. However, there are many questions that need to be answered and areas of the manuscript need to be improved to provide sufficient information for policymakers. I strongly recommend this manuscript be reviewed by a specialist in health economy before making any decision. The manuscript needs to be edited for language. I have provided detailed comments in the attached document.

Response:

We are thankful for your detailed comments on the content of the manuscript. We have made necessary changes in the manuscript as suggested by the reviewer. All the comments have been extracted from the attached document and responses have been summarised below:

Specific comments

1. Title:

a) Line no. 80-81; This can increase stigma related to HCV. Please change to "people infected with Hepatitis C virus" through the manuscript.

Response:

We thank the reviewer for making this observation. In view of this, we have revised the title as "Cost-effectiveness and budget impact analysis of facility-based screening and treatment of Hepatitis C in Punjab state of India."

Page no. 1, Line no. 1-2; Page no.3, Line no. 80-81.

2. Abstract:

a) Line no. 89-92; "scenario I – screening with Enzyme Linked Immunosorbent Assay (ELISA) followed by confirmatory HCV-Ribonucleic Acid (RNA) quantification and scenario II – screening with Rapid diagnostic test (RDT) kit followed by confirmatory HCV-RNA quantification." Do these two scenarios include treatment?

Response:

We thank the reviewer for seeking clarification on this. We would like to clarify that both the scenario I and II include treatment. We have made changes in the abstract as well as the text to make it more explicit.

Page no. 3, Line no. 96-100

b) Line no. 89; "No universal screening and treatment (routine care)" Is this routine care in Punjab? See my comment in the main text methods section.

Response:

In view of the reviewer's comment, we would like to clarify that this represents the routine scenario in Punjab where in the absence of an active screening program, care is provided on ad-hoc basis as and when patient presents to the health facility.

c) Line no. 95; "Both scenarios I and II dominate over no-screening policy." This is more for conclusion section.

Response:

We thank the reviewer for making this observation. We have shifted this to the conclusion as suggested by the reviewer. Page no. 3, line no. 109-110.

d) Line no. 98-99; "Screening the age cohort of 40-45 years at a coverage of 30%, with ELISA and RDT, would cost 209 and 190 million rupees, respectively." Do we need to compare this with the screening of whole population? If yes, we need to see that results here.

Response:

We would like to mention that our analysis also compares the age group screening (40-45 years) with screening all 18 years and above. Thus, we mentioned this in our results.

As suggested by the reviewer, we have revised the text to include the comparative results.

Page 3, Line no. 106-108.

e) Line no. 100; "One-time universal screening followed by HCV treatment" This is not mentioned in scenarios I and II in the abstract.

Response:

In view of the reviewer's comment, we have revised the text to make the statement clear.

Page no. 3, Line no. 96-99

f) Line no. 101; "However, budget impact of screening activity seems unsustainable." How do we know this? because it is expensive? Is this mentioned in the Result section of abstract?

Response:

We thank the reviewer for seeking clarification on this. We would like to state that this statement implies that screening activity poses a huge budget impact, specifically, when screening all people 18 years or older. We have mentioned this in the results section of the abstract as suggested by the reviewer.

Page 3, Line no. 104-106.

Also, we have provided a detailed explanation for the same, in this document, under response to comment (b) of the discussion section as well as in the manuscript; page no.17, Line no. 433-449.

g) Line no. 102-104; "we recommend beginning with screening the age cohort with RDT around mean age of disease presentation instead of whole population." What is this mean age? Do you mean "mean age" of people presenting at hospital for HCV treatment?

Response:

We thank the reviewer for seeking clarification on this. The given statement corresponds to the mean age at which HCV infected people present to the hospital to seek care, which is around 40-45 years of age.

3. Strengths and Limitations:

a) Line no. 125-126; "However, we varied the coverage estimate in our PSA from 10 to 50%. Also, we conducted a univariate analysis to report ICER/QALY at different coverage levels for 10 to 100%." Do these mean that assuming 30% coverage was not a problem? If yes, then why it is a limitation?

Response:

We thank you for seeking clarification on this point.

Coverage of screening is a significant determinant of cost-effectiveness as well as budget impact, therefore, it is important for policy makers and program managers to understand the impact of

coverage. As we did not have exact estimate for this, we made an assumption regarding the coverage of screening, therefore we mentioned this as a limitation of our analysis. However, tried to address this limitation by varying the coverage parameter in the sensitivity analysis. However, we understand and agree to the reviewer's suggestion. Thus, we have deleted this from our limitation section.

b) Line no. 127-128; "Secondly, high risk group (HRG) screening assessment would have been a better strategy to generate evidence as India is a resource limited setting and prevalence among HRGs is high." This is not a limitation of your analysis. It seems to be more discussion.

Response:

We thank the reviewer for making this observation. In view of the reviewer's comment, we deleted this from our limitations.

4. Introduction:

a) Line no. 141; "Though the national prevalence in India ranges from 0.9-1.9%". Is this prevalence of chronic HCV?

Response:

We thank the reviewer for seeking clarification on this. We would like to mention that this figure represents the prevalence of chronic HCV in India.

b) Line no. 149-151; "The high prevalence in Punjab can be attributed to the higher prevalence of unsafe medical practices and rather than due to intravenous drug use (IDU), therefore posing a high risk of reinfection as well." More information is needed to be reported here to understand the epidemiology of HCV in Punjab. Is unsafe medical practice still the lead cause? Prevalence of medical practices does not seem to be correct. Consider changing it.

Response:

We thank the reviewer for seeking clarification on this. In view of this, we have revised the text as follows:

"The high prevalence in Punjab can be attributed to the higher prevalence of unsafe medical procedures and blood transfusions. Another contributor to high HCV burden in Punjab is the Intravenous drug user (IDU) population, which is at a higher risk for reinfection as well."

Page 5, Line no. 154-157.

c) Line no. 154-155; "In 2016, Punjab government launched the "Mukh Mantri Punjab Hepatitis C Relief Fund (MMPHCRF) for free of charge universal treatment of HCV patients with all oral DAAs." Are all treatment services free or only DAAs?

Response:

We thank the reviewer for seeking clarification on this. We would like to mention that free treatment included DAAs as well as diagnostic as well as monitoring tests required during the treatment.

d) Line no. 158-160; "With the success of the Punjab Model and in concurrence with the global effort to eliminate HCV infection, India launched the National Viral Hepatitis Control Programme (NVHCP) in July 2018 which aims at improving access to diagnosis and treatment for viral hepatitis." How do we know it was successful? What is the definition of successful program here? Information about this program is needed?

Response:

We thank the reviewer for seeking clarification on this.

In view of this we have revised the text as follow:

"With the success of the Punjab Model, where a cure rate of more than 90% was achieved among those enrolled, and in concurrence with the global effort to eliminate HCV infection, India launched the National Viral Hepatitis Control Programme (NVHCP) in July 2018 which aims at improving access to

diagnosis and treatment for viral hepatitis. This program covers the entire gamut of Hepatitis including type A, B, C, D & E with a whole range of prevention, detection and treatment.”

Page 6, Line no. 165-170.

e) Line no. 167-169; “We undertook this analysis to assess the incremental cost per quality adjusted life year (QALY) for one-time universal screening followed by treatment of HCV infected population as compared to no screening in Punjab.” The provided background information does not support on-time universal screening. Based on the provided background information there are two options including screening the high-risk groups and screening people at age 40.

Response:

In light of the reviewers’ comment, we have revised the text to make the research question more explicit.

Changes made on Page no. 5, Line no. 153-158; Page no. 6, Line no. 171-180.

f) Line no. 169-171; “This study also aims to inform on which screening technology represents value for money and would yield a better financial sustainability for population-based screening in Punjab.” What does financial sustainability” mean here?

Response:

We thank the reviewer for making this observation.

Financial stability refers to the operational feasibility for the HCV screening program given the available fiscal space for funding it. To measure this, we estimated the impact of providing HCV screening on the health budget of Punjab state.

In light of the reviewers’ comment, we have revised the language of the text to explain the same.

Changes made on Page 6, Line no. 184-186.

5. Methods:

a) Line no. 187-188; “All the future costs and consequences are discounted at a rate of 3%.” What are the further consequences? Maybe listing them here!

Response:

We thank the reviewer for seeking this clarification.

In our analysis, future consequences are measured in terms of early vs late detection of HCV, new HCV infections averted and HCV deaths averted.

Changes made on Page no. 7, line no. 203-205.

b) Line no. 207; “A loss to follow up of 10% has been assumed at each step from testing to treatment.” Have you considered lost to follow up during treatment? non adherence to treatment? When treatment is offered to all patients these can be higher than routine care.

Response:

We thank the reviewer for seeking this clarification.

We would like to mention that we have considered all these elements. The efficacy estimates for the treatment, i.e, SVR rates, used in our analysis are based on analysis of MMPHCRF data for 48,088 HCV patients which was done using standard methods along with adjusting for loss to follow-up, drop outs, treatment failures and deaths.

c) Line no. 210; “Screening with RDT – Scenario II” It needs to be clear if this scenario includes treatment.

Response:

In view of the reviewer’s comment, we have made changes to specify both screening and treatment in description of scenario 1 and 2. Page no. 3, line no. 96-100; Page no. 8, line no. 224-227, 232-234.

d) Line no. 226-228; “A Markov model, with similar transition states, used in a previous analysis to assess the cost effectiveness of using velpatasvir (VEL) for treatment of HCV, was constructed to

simulate disease progression among the HCV population.” Is this to provide rational why Markov model was used in this analysis? If yes, how these studies can support the model that you have used? I can see you have referred to reference 38 and 39 for transition probability. Is this different from that?

Response:

We thank the reviewer to highlight this issue. The study mentioned here was done to assess the cost-effectiveness analysis of using Velpatasvir for HCV treatment. This study was done by the same group of authors as of this study and all our input parameters for this CEA are consistent with our previous analysis. The references mentioned here are the original sources of transition probabilities used in our previous analysis as well as this analysis.

e) Line no. 248-250; “Treatment associated costs include cost of out-patient (OPD) consultation for therapeutic purpose, in-patient care for hospitalisation which is required in advanced stages as well as cost of management of complications.” How about HBV and HIV tests for confirmed cases? How about SVR test? How about HCC follow up for people with cirrhosis.

Response:

We thank the reviewer for seeking clarification on this. We would like to mention that we did not include testing for HBV and HIV among those tested for HCV. In view of this, we have mentioned this in our limitations section.

Secondly, the cost of SVR test and other diagnostic tests as well as cost of managing each stage of HCV including the associated complications have been included. We have revised the language to make it more explicit.

Changes made on Page no. 10; line no. 270-276.

f) Line no. 261-262; “The cost of DAAs (ranging from ₹ 4000-17000, US \$ 57-242 per patient) were also as per the procurement prices in Punjab.” Is the cost of Ribavirin also included?

Response:

We thank the reviewer for seeking clarification on this. We would like to mention that the price of Ribavirin is also included. We have modified the text to include this.

Changes made on Page no. 11; Page no. 285-286.

g) Line no. 280-281; “The estimates for SVR rates were obtained from MMPHCRF data and are consistent with the estimates used in the previous analysis.” Why is this important?

Response:

In view of the reviewer’s comment, we have deleted this part from the text.

Changes made on Page no. 11, Line no. 306.

h) Line no. 314; “routine care, i.e, no-screening and treatment.” This is different from what has been mentioned in lines 194-198 as a definition of routine care. If you have considered no screening not treatment as baseline scenario it needs to be mentioned clearly. The current routine care as has been mentioned in lines 194-198 is provided treatment but not universal screening.

Response:

In light of reviewers’ comment, we have explained routine care at Page no. 7-8, Line no. 210-220 and maintained consistency throughout the manuscript.

i) Line no. 331; “cost of activities included in BIA.” How about cost of designing and implementing screening system, data collection and follow up. Cost for identifying and contacting people with positive screening test and positive HCV RNA.

Response:

We thank the reviewer for seeking clarification on this. We would like to mention that, as written in the text (Page no. 7, Line no. 192-194), screening will be delivered as a facility-based intervention at existing district hospitals which act as integrated testing and treatment sites, through which

MMPHCRF was operating. Thus, we did not include any additional costs for infrastructure. However, we used apportioned cost dedicated to HCV for each of the resources, i.e, building, space, equipment, consumable and other non-consumables. We did not apply any cost for identifying people as the program operated as a model of facility-based service wherein the patients would return to the facility to obtain their reports. Secondly, since there is no existing screening program in practice, we could not obtain this cost. In view of this, we included a 10% loss to follow up at each step from screening to treatment, so as to provide realistic estimates of costs and outcomes. Further, in consultation with the program experts, we have included the cost of information-education-counselling (IEC) and behaviour change communication (BCC) activities that will be conducted to motivated people for follow-up. Also, we included the cost of integrated training sessions that will be delivered to the staff before the initiation of screening activity.

6. Results:

a) Line no. 363-364; "When compared with no screening, both scenarios, i.e screening with ELISA as well RDT, dominate over no screening." The name for the two scenarios should be constant throughout the text.

Response:

We thank the reviewer for highlighting this. In view of this, we have made change made on Page no. 14, Line no. 373-377.

b) Line no. 365-366; "However, when the incremental gains of scenario I and II (relative to no screening) are compared, there is no statistically significant difference in outcomes as well as costs." What does "statistically significant difference" mean? Is this based on a statistical test?

Response:

We thank the reviewer for seeking clarification on this.

We would like to mention that when probabilistic sensitivity analysis (PSA) was conducted for both scenario I and II in comparison to routine scenario, there was an overlap between the confidence intervals of the incremental gains for both costs and outcomes (please see Table 2), which implies that the difference is statistically insignificant.

c) Line no. 374-378; "Results > Budget Impact Analysis" Here we need to see a summary of results. All the things have been written here are belong to Methods section.

Response:

We thank the reviewer for highlighting this. We have revised this as suggested by the reviewer under the results section (budget impact analysis).

Changes made on Page no. 16, Line no. 404-413.

7. Discussion:

a) Line no. 398-401; "Screening the population of 18 years and above, at base coverage of 30%, ELISA costs 1779

million, i.e, 3.8% of the Punjab state health budget whereas with RDT 1586 million rupees, i.e, 3.4% of the state health budget. However, if screening the age cohort of 40-45 years, with ELISA and RDT, account for 0.45% and 0.41% of the state health budget respectively." These are results and belong to results section. Here the main point should be provided and then discuss why is the point true.

Response:

We thank the reviewer for highlighting this. As suggested by the reviewer, we moved this to results section under the findings of budget impact analysis.

b) Line no. 403-404; "As the figures suggest, though the intervention is cost saving, but it is not feasible to initiate screening at full coverage and for the whole population." It is not clear which figures are referred to here. Why is the intervention not feasible? Because of cost or other factors?

Response:

We thank the reviewer for seeking clarification on this.

We would like to mention that the intervention is associated with very high budget impact given the available resources. BIA are financial tools which help us to estimate the future spending required in the chosen time horizon. In light of the results of BIA, one-time screening of 18 years and older population would cost in the range of 3.4% (RDT, scenario II) – 3.8% (ELISA, scenario I) of the state health budget, whereas, screening the population in the age-group of 40-45 years old will require spending 0.41% (RDT, scenario II) - 0.45% (ELISA, scenario I) of the state health budget. One way to assess whether the additional budget for health will be able to accommodate for higher allocation to any program is the analysis of the fiscal space. Fiscal space is dependent on: the macroeconomic factors for which the impact on the GDP is assessed. Secondly, one can generate the estimates for fiscal space through budget reprioritization. Thirdly, we can measure efficiency and estimate monetary value of efficiency enhancing measures. Consequent to this, we compare the required budget for screening with average annual growth in the health budget of Punjab state, which has been reported to 10%.⁵⁶ According to the India State-Level Disease Burden Initiative, cirrhosis and other chronic liver diseases account for 1.44% of total morbidity burden in Punjab.⁵⁷ Given this scenario, spending more than 0.144% of the health budget for HCV program seems unrealistic. Thus, opting for screening the age-group of 40-45 years with RDT instead of all 18 years and older appears operationally feasible.

We have added this in the discussion section as well. Changes made on Page no.17, Line no. 433-449.

c) Line no. 408-410; “if one dedicated person to HCV screening is deployed at each of the districts, the human resources will be sufficient to screen the age cohort of 40-45 years whereas, additional 1015 persons would be required to screen everyone 18 years and older.” Is this part of results of this analysis? Could you please clarify how did you find this. What is the target population in each district and what is the period of screening?

Response:

We thank the reviewer for seeking clarification on this. As mentioned, we considered screening as a one-time activity. We considered two scenarios: screening 18 years and older & screening around the age of disease presentation, i.e 40-45 years. According to the age criteria, we estimated the population eligible for screening (based on demographic data of Punjab state) in the two scenarios. We then estimated the requirement of human resource for one-time screening activity in both the scenarios, given the current resources. Currently, there are 22 district labs which conduct the test and 1 person in each lab is responsible for conducting HCV test in addition to conducting other tests as well. From the data of district laboratory in Punjab, we calculated the manhours spent by this person to conduct ‘n’ number of tests in a year. Further, we estimated the actual required manhours to conduct the screening for both mentioned scenarios at base coverage, i.e, 30%. This is how we calculated the require human resources.

Since there is no dedicated worker for HCV screening, we also estimated the requirement of human resources if one person is dedicated to conduct only HCV tests throughout. In such a scenario, if a screening program is initiated and one dedicated worker is deployed for HCV screening, it is possible to screen the entire cohort of people aged 40-45 years in one year at the base coverage. However, if we want to screen 18 year and older, we would need 1015 additional persons.

d) Line no. 413-415; “The findings of the study demonstrate that treating HCV was highly cost-effective along with annual re-testing for reinfection being cost-effective, thus supporting a policy change towards frequent testing.” How does it relate to the findings of this analysis?

Response:

We thank the reviewer for highlighting this. In view of this, we have revised the text to delete this part.

e) Line 429-430; “Thus, all the parameters used in the model are estimates from the real-world setting for which the analysis is being conducted.” Does it include expert opinion?

Response:

We thank the reviewer for seeking clarification on this.

Our analysis is based on the primary data collected for 48,808 patients of HCV in Punjab state. In addition, our estimates of quality of life are also derived from a pool of patients visiting/admitted to a tertiary care hospital which caters to the healthcare needs of North India. Thirdly, the cost data has been derived from the National Health System Cost database for India. Fourthly, the clinical data for parameters such as number of visit/follow ups/episodes of complications and their management was based on expert opinion.

f) Line no. 436-437; “Secondly, high risk group (HRG) screening assessment would have been a better strategy to generate evidence as India is a resource limited setting and prevalence among HRGs is high.” Is this your analysis limitation? High risk group need to be defined. Based on the methods of HCV transmission in Punjab.

Response:

We thank the reviewer for seeking clarification on this.

We would like to mention that we discussed this as a limitation because, given the huge population and limited availability of resources, one should focus on identifying the high-risk groups. However, we understand the reviewer’s concern that this could more of a discussion point. Thus, in view of this, we have deleted this from our limitation section.

Secondly, as suggested by the reviewer, we have defined HRG on Page 6, line no. 172-176.

g) Can model uncertainty and parameter uncertainty limit your findings?

Response:

We thank the reviewer for seeking clarification on this.

Both structural and parameter uncertainty are bound to impact the results of any model-based analysis. However, we conducted both one-way sensitivity analysis and PSA to address the same.

8. Conclusion:

Line no. 455-457; “We cannot deny that screening is important as less than 1% people in India are aware of their HCV status. Due to this, the needy population is unable to access the benefit of available low cost and highly efficacious drug treatment.” This is not from your results then should not be in conclusion.

Response:

We thank the reviewer for making this observation. In view of this, we have revised the conclusion as suggested by the reviewer.

9. Table 1: Demographic, epidemiologic, effectiveness and cost-related parameters; “Coverage parameters”. Check if these are correct.

Response:

We thank the reviewer for highlighting this. We have corrected this in light of the reviewer’s comment.

VERSION 2 – REVIEW

REVIEWER	Deuffic-Burban Sylvie Inserm, France I report having received consultancy honoraria from Intercept.
REVIEW RETURNED	17-Nov-2020
GENERAL COMMENTS	The authors revised the submitted manuscript based on reviewers' comments. In particular, I thank the authors for responding satisfactorily to my specific comments.

REVIEWER	Davoud Pourmarzi Australian National University, Australia
REVIEW RETURNED	30-Nov-2020

GENERAL COMMENTS	I would like to congratulate the authors on their work. My questions are answered and comments are addressed. However, I have provided some minor comments in the attached file for improving the clarity of the manuscript. - The reviewer provided a marked copy with additional comments. Please contact the publisher for full details.
---

VERSION 2 – AUTHOR RESPONSE

Reviewer(s)' Comments to Author:

Reviewer#1

The authors revised the submitted manuscript based on reviewers' comments. In particular, I thank the authors for responding satisfactorily to my specific comments.

Response:

We are extremely thankful to the reviewer for the comments and inputs which helped us improve and strengthen the content of the manuscript.

Reviewer#2

General comments

I would like to congratulate the authors on their work. My questions are answered and comments are addressed. However, I have provided some minor comments in the attached file for improving the clarity of the manuscript.

Response:

We are highly grateful to the reviewer for the comments and suggestions to improve and strengthen the content of the manuscript. We have incorporated the suggestions provided by the reviewer in our manuscript and the changes have been summarised below according to each comment as it appeared in the attached pdf.

Page 5, Line 157. Use another term. Maybe "the high percentage of unsafe...."

Use of term "Prevalence" for unsafe medical procedures and blood transfusion doesn't make sense.

Response:

We thank the reviewer for highlighting this.

We have revised this sentence as “The high prevalence in Punjab can be attributed to the high percentage of unsafe medical procedures and blood transfusions.”

Changes made on Page 5, Line 154-155.

Page 5, Line 157; Page 6, Line 168, 172, 178, 180. Provide references.

Response:

In view of the reviewer’s comment, we have added references for all the mentioned lines.

Page 5, Line 158. You are trying to present reasons why HCV prevalence in Punjab is higher than other states, then readers need see some data that shows high risk practices and behaviours in Punjab is more common than in other state, if these are really the reasons.

Response:

We thank the reviewer for seeking clarification on this.

In view of this, we have revised the text as “Furthermore, Punjab ranks 2nd amongst the top 10 Indian states with highest number of people who inject drugs (PWID), which is a contributor to increased risk for reinfection as well.”

Changes made on Page 5, Line 156-157.

Page 5, Line 162. “...free treatment included DAAs as well as diagnostic as well as monitoring tests required during the treatment.” These should be clearly mentioned. Treatment may or may not include diagnostic and monitoring tests and examinations.

Response:

We thank the reviewer for highlighting this. We have revised the text as “In 2016, Punjab government launched the “Mukh Mantri Punjab Hepatitis C Relief Fund (MMPHCRF)” for free treatment including DAAs as well as diagnostic as well as monitoring tests required during the treatment of HCV patients.”

Changes made on Page 5, Line 159-161.

Page 8, Line 228. You should consider that lost to follow up and adherence to treatment among patients who are diagnosed by universal screening are different from patients who are seeking for treatment. This should be mentioned in the limitation of this study that you assumed the same lost to follow up and adherence for patients diagnosed through universal screening.

Response:

We thank the reviewer for seeking clarification on this. We agree with the reviewer on that lost to follow up and adherence may be different for patients diagnosed by universal screening and among those who are seeking for treatment. We have also mentioned this in our limitations on Page 19, Line 481-484.

However, we would like to state that these estimates are unlikely to alter the direction of results as the proportion of people seeking diagnostic and treatment services for HCV on their own is very low.

Page 8, Line 250-252. Based on the authors' response to my question on previous version, this part seems extra.

Response:

In view of the reviewer's suggestion, we have removed this part from the text of the manuscript.

Page 15, Line 391-392. Your respond to my question in the previous version of manuscript should be clearly mentioned in the methods section to help readers understand what has been defined as "statistically significant difference".

Response:

We thank the reviewer for highlighting this. In view of this, we have revised the text as "However, when the incremental gains of scenario I and II (relative to no screening) are compared, there was an overlap between the confidence intervals of the incremental gains for both costs and outcomes which implies that the difference is statistically insignificant."

Changes made on Page 15, Line 386-389.

Page 19, Line 482-483. HBV and HIV are needed for starting treatment for people with chronic HCV not for all people testing for HCV.

Response:

We thank the reviewer for highlighting this. We have revised the text as "We would also like to mention that we did not include testing for HBV and HIV among those tested who tested positive for HCV."

Changes made Page 19, Line 479-480.